# Evaluation of Haptoglobin and Its Proteoforms as Glioblastoma Markers

**DOI:** 10.3390/ijms22126533

**Published:** 2021-06-18

**Authors:** Stanislav Naryzhny, Natalia Ronzhina, Elena Zorina, Fedor Kabachenko, Maria Zavialova, Viktor Zgoda, Nikolai Klopov, Olga Legina, Rimma Pantina

**Affiliations:** 1Institute of Biomedical Chemistry, Pogodinskaya, 10, 119121 Moscow, Russia; el.petrenko@bk.ru (E.Z.); mariag.zavyalova@gmail.com (M.Z.); vic@ibmh.msk.su (V.Z.); 2National Research Center “Kurchatov Institute”, Petersburg Nuclear Physics Institute, 188300 Gatchina, Russia; ronzhina@list.ru (N.R.); klopov_nv@pnpi.nrcki.ru (N.K.); olgaleg@mail.ru (O.L.); pantina61@mail.ru (R.P.); 3Peter the Great St. Petersburg Polytechnic University, 195251 St. Petersburg, Russia; fkabachenko@gmail.com

**Keywords:** haptoglobin, biomarker, zonulin, glioblastoma

## Abstract

Haptoglobin (Hp) is a blood plasma glycoprotein that plays a critical role in tissue protection and the prevention of oxidative damage. Haptoglobin is an acute-phase protein, its concentration in plasma changes in pathology, and the test for its concentration is part of normal clinical practice. Haptoglobin is a conservative protein and is the subject of research as a potential biomarker of many diseases, including malignant neoplasms. The Human *Hp* gene is polymorphic and controls the synthesis of three major phenotypes—homozygous Hp1-1 and Hp2-2, and heterozygous Hp2-1, determined by a combination of allelic variants that are inherited. Numerous studies indicate that the phenotype of haptoglobin can be used to judge the individual’s predisposition to various diseases. In addition, Hp undergoes various post-translational modifications (PTMs). Glioblastoma multiform (GBM) is the most malignant primary brain tumor. In our study, we have analyzed the state of Hp proteoforms in plasma and cells using 1D (SDS-PAGE) and 2D electrophoresis (2DE) with the following mass spectrometry (LC ES-MS/MS) or Western blotting. We found that the levels of α2- and β-chain proteoforms are up-regulated in the plasma of GBM patients. An unprocessed form of Hp2-2 (PreHp2-2, zonulin) with unusual biophysical parameters (pI/*M*w) was also detected in the plasma of GBM patients and glioblastoma cells. Altogether, this data shows the possibility to use proteoforms of haptoglobin as a potential GBM-specific plasma biomarker.

## 1. Introduction

High-grade glioma (GBM, glioblastoma multiform) is the most common brain tumor with an extremely poor prognosis. Its malignancy makes GBM the fourth biggest cause of cancer death [1,2]. Today, the main methods of diagnosis of this disease are computed tomography and brain biopsy. Usual GBM treatment combines maximal resection with radiotherapy and adjuvant therapies [3]. Despite this radical approach, most patients suffer recurrence because of GBM heterogeneity. The average survival period is ~15 months, pointing to the need to find the potential predictive diagnostic, and prognostic biomarkers. Combined efforts of physicians and scientists were applied in this direction, where tissue samples or non-solid biological liquids were analyzed as a source of these potential biomarkers [4]. The testing of these fluids is defined as a liquid biopsy [5]. Usually, liquid biopsy is carried out by using blood but other biofluids, such as saliva or urine, can be also used. Cerebrospinal fluid (CSF) has also been used in brain tumor studies, but it requires an invasive lumbar puncture procedure [6]. Blood sampling as the most popular liquid biopsy is a minimally invasive and preferred procedure that could provide information about tumors [7]. GBM induces such a microenvironment that allows the establishment of a more permeable blood–brain barrier (BBB) [8]. GBM-specific content can go through BBB and be tested in circulating blood [3,9]. The development of GBM signatures through protein biomarker profiling is a high priority here. Qualitative and quantitative changes in the specific protein groups could serve as an indicator of GBM. Such profiling not only may give a clue regarding tumor classification, but may identify clinical biomarkers or targets for the development of clinical treatments [10,11]. In our previous experiments, we tested GBM cell lines to obtain proteomics information specific to this disease [12]. A comprehensive 2DE map of glioblastoma cell proteins and a list of proteins overexpressed in glioblastoma cells as “potential glioblastoma biomarkers” were generated [12,13,14,15].

In our present work, we decided to check the possibility of liquid biopsy and explore aspects of plasma proteoforms heterogeneity as potential GBM biomarkers. The data obtained pointed to haptoglobin as a protein, which proteoform profile could be a promising biomarker of GBM. This idea is also supported by a line of publications [16,17,18,19,20,21], where the possible role of haptoglobin as an oncomarker is discussed. Hp is one of the abundant proteins (0.38–2.08 g/L) in plasma that is the hemoglobin-binding an acute-phase protein. Haptoglobin is composed of two types of polypeptide chains (α and β) covalently linked by disulfide bonds. Both chains are encoded by a single gene located on chromosome 16 [11,22]. Haptoglobin has several unique biophysical characteristics. Only humans have the polymorphic *Hp* gene that has three structural alleles that control the synthesis of three major phenotypes of haptoglobin, homozygous Hp1-1 and Hp2-2, and heterozygous Hp2-1, determined by a combination of allelic variants (α1 or α2) that are inherited. Additionally, Hp undergoes various post-translational modifications (PTM). These are structural transformations (removal of the signal peptide, cutting off the Pre-Hp precursor molecule into two subunits, α and β, limited proteolysis of α-chains, formation of disulfide bonds, multimerization), as well as chemical modifications of α- and β-chains. Especially many protein variants (proteoforms) are formed during glycosylation of the β-chain of Hp at four Asn sites (Asn184, Asn207, Asn211, Asn241) [23,24]. Using 2DE, these proteoforms are detected as a train of at least 10 spots. The proteoforms of Hp α2-chain are known to migrate as at least three spots with similar masses but different pI, whereas α1-chain-as up to three main spots [25]. In some cases, the Pre-Hp precursor (more exactly, only Hp2-2 type) can function under the name zonulin as a single polypeptide performing differently from haptoglobin’s functions [26]. Zonulin is a protein that is involved in the regulation of the spaces between epithelial cells. Zonulin level in blood reflects intestinal permeability, and its increased levels are considered to be a marker of impaired intestinal barrier [26]. However, zonulin is secreted not only by enterocytes; it has been described in several other tissues, e.g., brain, heart, liver, lungs, kidney, and skin [26,27]. Thus, the levels of zonulin in plasma may do not only reflect intestinal secretion, but also secretion from other organs [28]. In our study, we have analyzed proteoforms of haptoglobin in more detail. It was revealed that α-chain and especially β-chain have more complicated sets of proteoforms than was thought before. We found that the levels of α2- and β-chain proteoforms are up-regulated in the plasma of GBM patients compared to healthy donors. Additionally, the unprocessed form of haptoglobin, Hp2-2 (PreHp2-2, zonulin), was detected in samples from GBM patients, not from other donors.

## 2. Results

To obtain more information about plasma proteomes in connection to glioblastoma we have carried out a comparative proteomic analysis of plasma samples from GBM patients and other donors. A gel-based proteomic analysis using SDS-PAGE (1DE), 2DE, and sectional 2DE techniques was performed in combination with MS and immunodetection (Western blotting).

### 2.1. Detection of Differentially Expressed Proteins in GBM Plasma Samples versus Non-Tumorous Plasma Samples Based on SDS-PAGE

Initially, we have performed a comparative analysis of different groups of plasma samples using SDS-PAGE. These groups were: (a) GBM (*n* = 22), (b) other cancers (*n* = 7), (c) acute-phase, non-related with cancer (*n* = 6), (d) healthy group (*n* = 16). The equal amounts of proteins from each sample were loaded, and after electrophoresis, the gels were stained with Coomassie R350. It should be mentioned that among all samples, it was four samples (two from the healthy group, one from other cancers, and one from the GBM group) where α2-chain was absent. That means these persons have an Hp1-1 phenotype, and for statistical analysis of α2-chain abundancy, these samples were not included. Talking about Hp phenotypes in all samples, the main one was Hp2-2, a few Hp2-1, and only 2—Hp1-1. The representative gels of such analysis are shown in Figure 1a–c. Densitometry of each line was performed, and the intensity of every band was measured. Among different bands, the bands that correspond to *M*w ~35,000 and ~18,000 were overexpressed in GBM samples. Western blot analysis and two-dimensional gel electrophoresis (2DE) revealed that these bands correspond to Hp β-chain and α2-chain (Figure 2a–c).

According to densitometry-based statistical analysis, a level of α2- and β-chains is increased in acute-phase samples. It is no wonder, as Hp is an acute-phase protein. In the cancer-specific samples, the Hp level is even higher and is highest in GBM samples (Figure 1d,e). Interestingly, comparative analysis based on densitometry of Coomassie-stained gels or immune-stained membranes gave similar results (Figure 1d,e and Figure 2d,e). It means that for α2-chain and β-chain, we can detect their amounts in SDS-PAGE just after Coomassie staining. According to the known amount of loaded protein, we can calculate concentrations of α2- and β-chain of Hp in the samples. This data shows that using just Coomassie staining of SDS-PAGE gels we can evaluate the level of Hp α2- chain and β-chain in a sample.

A more detailed evaluation of Hp proteoforms was performed based on 2DE followed by shot-gun mass-spectrometry (ESI LC-MS/MS).

### 2.2. An Analysis Based on Spots in 2DE Gels

To obtain profiles (patterns) of Hp proteoforms, we performed 2DE separation of plasma proteins. Proteins after 2DE separation were stained with Coomassie and the produced spots, were analyzed by mass spectrometry (ESI LC-MS/MS analysis). Proteins located in many spots were identified. The obtained data are mainly following the 2DE plasma map deposited at the SWISS-2DPAGE database http://world-2dpage.expasy.org/swiss-2dpage/ (accessed on 11 April 2021) [29]. Additionally, we found that many spots contain several different proteins (Appendix A Appendix A). We have paid special attention to spots where Hp was detected (Hp spots). The proteoform clusters (patterns) of α2-chain and β-chain are well-represented in plasma (Figure 3).

We performed a comparative 2DE quantitation analysis of α2-chain and β-chain patterns in plasma of healthy people and GBM patients. Additionally, to obtain more reliable data about the presence of different proteoforms of Hp in plasma from healthy people or GBM patients, we applied Western blotting using Ab against α2-chain (Figure 4).

The 2DE-based analysis confirmed the results obtained using SDS-PAGE. The levels of proteoforms of Hp α2-chain and β-chain are elevated in the plasma of GBM patients. As Hp α2-chain has several proteoforms it is necessary to figure out which proteoform(s) could be responsible for Hp elevated level in plasma of GBM patients. According to data obtained, it looks like there is no preference between different proteoforms in this upregulation. In the case of α2-chain, three major proteoforms can be observed in Coomassie-stained 2DE gel (Figure 3). Comparative analysis using Coomassie spot absorbance or immunostaining (Western blotting) did not show preference of any spot for the elevated level of Hp α2-chain in plasma of GBM patients. What is more, the ratios in intensities of three spots (proteoforms) obtained using Coomassie staining, or immunostaining (Western blotting) were practically the same. It shows that the impurity of Hp spots by other proteins does not affect the assessment of Hp representation in these spots. A similar situation was observed in the case of major proteoforms of β-chain (data not shown). However, it should be mentioned that the spot-based MS-analysis also revealed more locations of Hp proteoforms than it was detected before (Appendix A Appendix A). The β-chain especially has many proteoforms. This data pointed to the possibility of detecting more Hp proteoforms if the whole gel will be analyzed. Therefore, we applied our 2DE sectional (pixel) analysis to plasma samples.

### 2.3. A Sectional 2DE Analysis

As haptoglobin was detected not only in “Hp spots”, but also in some spots that “are occupied” by other plasma proteins (Appendix A Appendix A), it means that there is a chance of detecting more Hp proteoforms when a whole gel analysis is performed. It can be done using our sectional (pixel) 2DE analysis [30], in which case, a whole 2DE gel is cut into 96 sections with determining coordinates (pI/*M*w), and each section is treated according to the protocol for shot-gun mass-spectrometry (ESI LC-MS/MS analysis). Finally, after analysis of mass-spectra, a list of proteins located in each section is obtained. If a protein is detected in different sections, it is considered that this protein has different proteoforms.

In the case of plasma proteins, it happened that Hp α-chain was detected at least in 10 sections and β-chain—at least in 30 sections of the 2DE gel (Figure 5, Appendix A Appendix A). Keeping in mind that sections are big enough to accommodate several different proteoforms, the real number of proteoforms is even bigger than the number of sections. For instance, in the case of β-chain, section C6 with coordinates: pI 5.11–5.80/*M*w 35,000−40,000 accommodates at least ten spots representing the β-chain of Hp that shown in Figure 2. This data shows that plasma Hp can exist in much more proteoforms than was reported previously [29]. What is especially interesting, this number is increased in the case of cancer (Figure 5).

Another aspect demanding attention is colocalization in some cases of α- and β-chains. Mainly, as expected, the patterns of Hp α- and β-chain proteoforms occupy different sections because of the big difference in *M*w (theoretical *M*w for α1-chain—9192, α2-chain—15,847, β-chain—27,265) of these polypeptides. Additionally, we see something special, when both α- and β-chains are detected in the same section. In this case, if a section corresponds to a polypeptide with *M*w bigger than 40,000 and contains both α2- and β-chains, it means that located here α2- and β-chains are covalently linked together. In other words, the polypeptide detected in this section represents an unprocessed Pre-Hp precursor (zonulin). The theoretical biophysical parameters of zonulin are pI 6.3/*M*w 43,000. Section D6 (pI 5.80–6.30/*M*w 35,000–40,000) is close to these parameters. What is more interesting, the similar situation we observed also in sections F6, G5, and G6, but only for the plasma of GBM patients (not in the control sample, or colon cancer sample) (Figure 5). As the pH range of these sections (8–10) is much higher than the theoretical pI 6.3, it means that zonulin detected in these sections is bearing some PTMs that are responsible for this shift.

As this form of zonulin was detected only in the plasma of GBM patients we also have checked the Hp pattern in glioblastoma cells (Figure 6a,c). Interestingly, two proteoforms were detected: one in section B2 (pI 4.45–5.11/*M*w 83,000–116,000) and another in section G5 (pI 7.82–8.86/*M*w 40,000–52,000).

What is interesting, a similar Hp pattern was obtained with extract of LEH (human lung embryonic fibroblasts) (Figure 6b,d, Appendix A Appendix A). It means that only full-length preHp2-2 (zonulin) presents in these cells. This information is in full concord with known data about the involvement of zonulin not only in intestinal permeability, but also in a wide range of extraintestinal epithelia as well as the ubiquitous vascular endothelium, including the BBB [32]. Zonulin can cross BBB, enter the bloodstream, and be detected in plasma.

## 3. Discussion

We have paid special attention to Hp using a panoramic analysis of plasma proteins. It should be noted that Hp is an interesting protein from many points of view. First, Hp is an active component of plasma and is involved in many processes that supporting human body homeostasis. Hp binds free hemoglobin (Hb) and plays a critical role in tissue protection and prevention of oxidative damage. Besides, it has some regulatory functions. Hp has special biophysical characteristics. The human *Hp* gene is polymorphic, and has three structural alleles that control the synthesis of three major phenotypes of haptoglobin: homozygous Hp1-1 and Hp2-2, and heterozygous Hp2-1, determined by a combination of allelic variants. Numerous studies indicate that the phenotype of haptoglobin can be used to judge the individual predisposition of a person to various diseases. Additionally, Hp undergoes various post-translational modifications (PTMs). These characteristics indicate the possibility of the existence of Hp in the form of a multitude of proteoforms, probably performing different functions. Recently, an unprocessed form of Hp (zonulin) started to attract attention as a biomarker. For example, it was reported that the Pre-Hp could be a marker of hepatoma [33]. Accordingly, the quantitative and qualitative changes in Hp proteoforms in connection with cancer, especially GBM, could be a valuable source of biomarkers.

Hp is an acute-phase protein, so, it is expected to find increased plasma levels of Hp in case of stress situation. Accordingly, we found that in plasma samples from such patients, the Hp level is markedly increased (Figure 1 and Figure 2). However, in the case of cancer (possibly the strongest stressful situation), the Hp level is increased, even more, reaching the highest value in the case of GBM (Figure 1 and Figure 2). Therefore, our data confirm the possibility to use an elevated level of Hp as a nonspecific GBM biomarker.

As far as the GBM-specific proteoforms of Hp, still, there is a chance to find it among multiple proteoforms of the β-chain (Figure 5c,f,i)). However, the situation here is complicated and needs a special analysis. The most promising data was obtained with Hp unprocessed form, Pre-Hp2 (zonulin). A comparative 2DE sectional analysis has revealed its presence only in GBM plasma samples (Figure 5). This is in agreement with information about zonulin expression by glioblastoma cells [32]. We also tested the glioblastoma cell line and found only zonulin, not processed forms of haptoglobin here (Figure 6a,c). It should be also mentioned that in all these samples, zonulin presents in alkaline form (pI~8) while its theoretical pI is 6.3. To make the situation more interesting, we should add that the same form of zonulin presents in lung cells (LEH) as well (Figure 6b,d). This is in agreement with publications, where the role of zonulin in invasiveness and vascularity of GBM is discussed [34].

Taken together, these facts can be preliminarily considered as a possibility to use all Hp proteoforms as a biomarker panel for GBM, where the level of α- and β-chain is a nonspecific marker of a tumor, and zonulin is a specific marker of GBM. However, the situation with zonulin should be analyzed in more detail. We need more statistically relevant information about its presence in the plasma of GBM patients. Additionally, PTMs that are responsible for the big shift in pI of this protein need to be disclosed. This is what we are planning to investigate in our further experiments.

## 4. Materials and Methods

### 4.1. Cells and Plasma

A glioma cell line (Glia-L) generated in the Laboratory of Cell Biology (NRC «Kurchatov Institute»-PNPI, Gatchina, Russia) or LEH (human embryonic lung fibroblasts) were cultured in an atmosphere of 5% CO_2_ at 37 °C in DMEM/F12 (1:1) medium containing L-glutamine (BioloT, Saint Petersburg, Russia) and supplemented with 5% fetal bovine serum [14,15].

The plasma collection protocol was approved by the local ethics rules of the Burdenko Research Institute of Neurosurgery (Moscow, Russia) [35]. All glioma patients enrolling in the study were diagnosed with a primary grade IV glioblastoma (GBM). Additionally, blood samples were collected in the Pavlov First Saint-Petersburg State Medical University from non-GBM patients and healthy donors. Informed consent was signed by all patients and donors. Venous blood samples were obtained after an overnight fast into EDTA tubes and centrifuged at 1500× *g* 10 min at room temperature. The plasma was stored at −80 °C in cryotubes until further processing.

### 4.2. Immunostaining (Western Blotting)

Proteins were transferred for 2 h at 28 V from the gel onto PVDF membrane (Hybond P, pore size 0.2 μm, GE Healthcare, Pittsburgh, PA, USA) by placing the gel and membrane between two sheets of thick transfer paper (Bio-Rad, Hercules, CA, USA), impregnated with a transfer buffer (48 mM Tris, 39 mM glycine, 0.037% SDS, 20% ethanol). After transfer, the membrane was treated according to a protocol of Blue Dry Western [36]: stained by 0,1% Coomassie R350, dried, and treated with antibodies [21]. Primary antibodies were mouse monoclonal anti-Hp (C8, sc-, or F8, sc-390962, from Santa Cruz Biotechnology, Santa Cruz, CA, USA) in dilution 1/25 (80 ng/mL in TBS [25 mM Tris (pH 7.5) and 150 mM NaCl] containing 3% [*w/v*] BSA). Secondary goat anti-mouse immunoglobulins G labeled by horseradish peroxidase (NA931V, “GE Healthcare”) were used in TBS containing 3% [*w/v*] nonfat dry milk (1/5000 dilution). The reaction was developed by ECL (Western Lightning Ultra, “PerkinElmer”, Waltham, MA, USA) and X-ray film (Amersham Hyper film ECL) upon exposure of 10 s to 30 min.

### 4.3. Sample Preparation and Two-Dimensional Electrophoresis (2DE)

Samples were prepared as described previously [30]. Cells (~10^7^) containing 2 mg of protein, were treated with 100 µL of lysis buffer (7 M urea, 2 M thiourea, 4% CHAPS, 1% dithiothreitol (DTT), 2% (*v*/*v*) ampholytes, pH 3–10, protease inhibitor cocktail). Proteins were separated by IEF using Immobiline DryStrip 3–11 NL, 7 cm (GE Healthcare) or 5–8, 7 cm (BioRad, Hercules, CA, USA) following the manufacturer’s protocol. The samples in the lysis buffer were mixed with rehydrating buffer (7 M urea, 2 M thiourea, 2% CHAPS, 0.3% DTT, 0.5% IPG (*v*/*v*) buffer, pH 3–11 NL, 0.001% bromophenol blue) in final volume of 125 µL (250–300 µg of protein)/strip. Strips were passively rehydrated for 4 h at 4 °C. IEF was performed on an Hoefer™ IEF100 (Thermo Fisher Scientific, Waltham, MA, USA), which was programmed as follows: first step—300 V 1 h, second step—gradient to 1000 V, 1 h, the third step—gradient to 5000 V, 1.5 h, fourth step—5000 V 1 h, temperature 20 °C and maintained at the voltage 500 V. After IEF, strips were soaked 10 min in the equilibration solution (50 mM Tris, pH 8.8, 6 M urea, 2% sodium dodecyl sulfate (SDS) and 30% (*v*/*v*) glycerol, 1% DTT). This process was followed by 10 min incubation in the equilibration solution containing iodoacetamide instead of DTT (50 mM Tris, pH 8.8, 6 M urea, 2% sodium dodecyl sulfate (SDS) and 30% (*v*/*v*) glycerol, 5% iodoacetamide). The strips were placed on the top of the 12% polyacrylamide gel of the second direction and sealed with a hot solution of 1 mL of 0.5% agarose prepared in electrode buffer (25 mM Tris, pH 8.3, 200 mM glycine, and 0.1% SDS) and electrophoresed to the second direction under denaturing conditions using the system Hoefer™ MiniVE Mini Vertical Electrophoresis Unit (GE Healthcare). Electrophoresis was carried out at room temperature at a constant power of 3.0 W per gel [35]. Gels were stained with Coomassie Blue R350, scanned by ImageScanner III (GE Healthcare), and analyzed using Image Master 2D Platinum 7.0 (GE Healthcare).

In the case of a semi-virtual 2DE, the 18 cm IPG strip was cut into 36 equal sections. For complete reduction, 300 µL of 3 mM DTT, 100 mM ammonium bicarbonate were added to each sample and incubated at 50 °C for 15 min. For alkylation, 20 µL of 100 mM iodoacetamide (IAM) were added to the same tube and incubated in the dark at room temperature for 15 min. For digestion, a stock solution of trypsin (0.1 mg/mL) was diluted 1:10 by 25 mM ammonium bicarbonate, and 100 µL of diluted trypsin was added into each tube. Samples were incubated overnight for 4–24 h at 37 °C. Supernatants that may contain peptides that have diffused out of the gel slices were collected into new labeled 0.5 mL tubes. Peptides were extracted by adding 150 µL of 60% acetonitrile, 0.1% trifluoroacetic acid (TFA) to each tube containing gel slices. Extracts were dried in Speed Vac, reconstituted in 20 µL of 0.1% TFA, and analyzed by Orbitrap Q-Exactive Plus mass spectrometer. Protein identification and relative quantification were performed using Mascot “2.4.1” (Matrix Science) and emPAI. A table with information about all detected protein proteoforms was built. All proteins detected in the same section were given the pI of this section. Accordingly, the same proteins detected in different sections were considered as different proteoforms.

### 4.4. ESI LC-MS/MS Analysis

All procedures were performed according to the protocol described previously. For the 2DE sectional analysis, the gel was divided into 96 sections with determined coordinates. Each section (~0.7 cm^2^) was shredded and treated by trypsin according to the protocol for single spot identification with proportionally increased volumes of solutions. Tryptic peptides were eluted from the gel with extraction solution (5% (*v*/*v*) ACN, 5% (*v*/*v*) formic acid) and dried in a vacuum centrifuge. Peptides were dissolved in 5% (*v*/*v*) formic acid. Tandem mass spectrometry analysis was carried out in duplicate on an Orbitrap Q-Exactive mass spectrometer (“Thermo Scientific”, Waltham, MA, USA). Mass spectra were acquired in positive ion mode. High-resolution data was acquired in the Orbitrap analyzer with a resolution of 30,000 (*m*/*z* 400) for MS and 7500 (*m*/*z* 400) for MS/MS scans.

The data were analyzed by SearchGui [37] using the following parameters: enzyme—trypsin; maximum of missed cleavage sites—2; fixed modifications—carbaidomethylation of cysteine; variable modifications—oxidation of methionine, phosphorylation of serine, threonine, tryptophan, acetylation of lysine; the range of the precursor mass error—20 ppm; the product mass error—0.01 Da. As a protein sequence database, NeXtProt (October 2014) was used. Only 100% confident results of protein identification with detection of at least two peptides were selected for presentation. Two unique peptides per protein were required for all protein identifications. Exponentially modified PAI (emPAI), the exponential form of protein abundance index (PAI) defined as the number of identified peptides divided by the number of theoretically observable tryptic peptides for each protein, was used to estimate protein abundance.

### 4.5. Statistical Analysis

The resulting images from SDS-PAGE gels, pre-digitized with a Bio-Rad ChemiDoc MP instrument, were processed in the Bio-Rad Image Lab software. To process 2DE images, the Progenesis SameSpots (Nonlinear Dynamics, Newcastle upon Tyne, UK) program was used. To visualize the obtained data, graphs were built using Python and the Matplotlib library. Data preparation was carried out using the Pandas library. The significance was accepted at *p* < 0.05. All quantitative data were recorded as median (SD). In the plots obtained, boxes with whiskers express the median, 25% and 75% quartiles, the statistically significant range of the sample (whiskers and caps), as well as outliers-circles.

## 5. Conclusions

The collected up-to-date information shows that the concentration of certain proteins in plasma including Hp may indicate the presence of GBM. Despite these findings, none of these proteins alone was sufficiently specific and sensitive to serve as a diagnostic marker. Thus far, no single molecular marker was found to follow the dynamics of glioblastoma and so do not have clinical value [38]. The lack of consistency is due to the disease and patient heterogeneity and involvement of these proteins in various physiological and pathological processes. Another important limitation is a wide range of Hp levels in normal serum. In the future, a signature of multiple biomarkers including Hp may prove to be an especially useful tool in glioma subtypes diagnosis that will improve information about prediction and prognosis. Additionally, the characterization of a Hp phenotype may increase the sensitivity and reliability of the screening biomarkers.

## Figures and Tables

**Figure 1 ijms-22-06533-f001:**
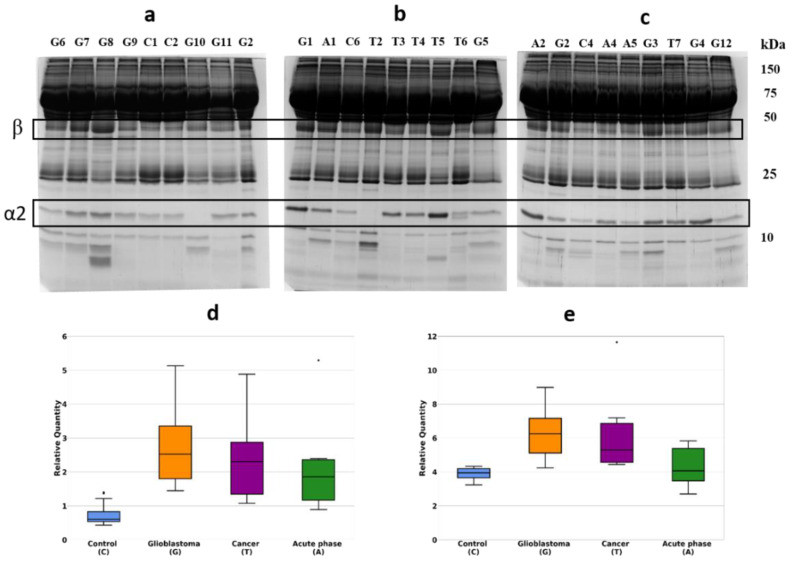
SDS-PAGE of plasma samples, Coomassie staining. The representative gels stained with Coomassie (**a**–**c**). Statistical analysis of the band intensity of α2-chain (**d**). Statistical analysis of the band intensity of β-chain (**e**). GBM samples are marked as G, control—as C, cancer—as T, acute phase—as A.

**Figure 2 ijms-22-06533-f002:**
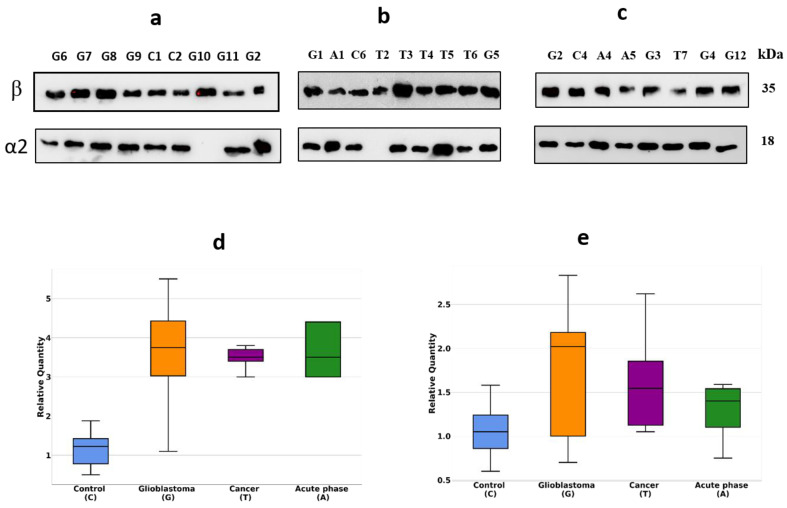
Western blotting of plasma samples. The duplicates of the gels shown in Figure 1 were tested using Ab against α2-chain and β-chain (**a**–**c**). Intensity statistical analysis of the bands of α2-chain and β-chain (**d**,**e**). GBM samples are marked as G, control—as C, cancer—as T, acute phase—as A.

**Figure 3 ijms-22-06533-f003:**
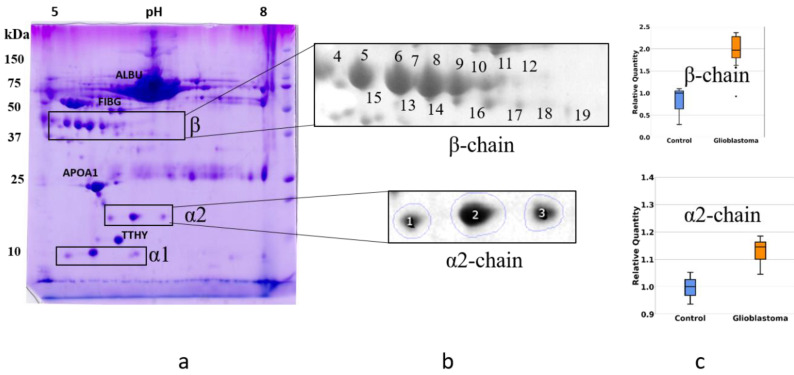
The comparative 2DE analysis of the abundance of Hp α2- and β-chain proteoforms in plasma of control and GBM patients. A representative Coomassie-stained 2DE gel of plasma proteins (GBM) with identified positions of α- and β-chains (**a**). The enlarged areas with spots containing α2- and β-chain proteoforms (**b**). MS data about proteins detected in different spots can be found in the Appendix A. A comparative statistical analysis of sum spot intensities in control and GBM samples (**c**).

**Figure 4 ijms-22-06533-f004:**
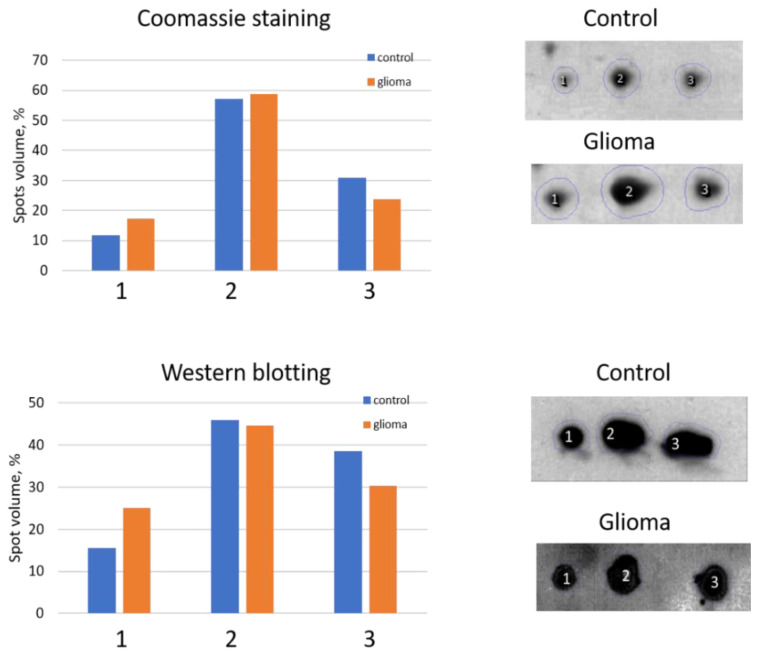
An analysis of three main spots of α2-chain. The relative intensity of each spot in control or GBM samples was estimated after Coomassie staining (**top**) or Western blotting (**bottom**). The spot numbers are the same as in Figure 3.

**Figure 5 ijms-22-06533-f005:**
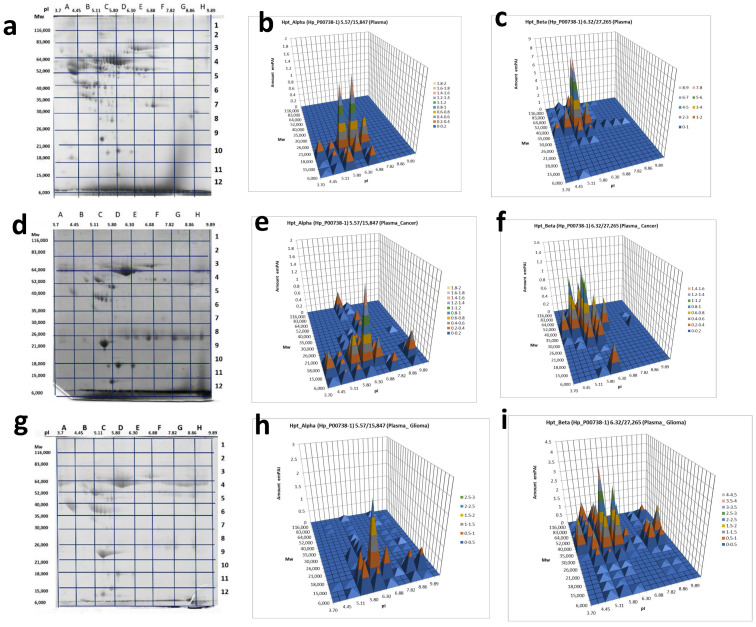
2DE sectional analysis of Hp proteoforms in plasma. (From left to right) Sections in 2DE gel selected for the following ESI LC−MS/MS analysis, Hp α-chain or β-chain detection in sections: (**a**–**c**) control, a pooled plasma of the mix of samples from 54 healthy donors [31], (**d**–**f**) a patient with colon cancer, and (**g**–**i**) a GBM patient.

**Figure 6 ijms-22-06533-f006:**
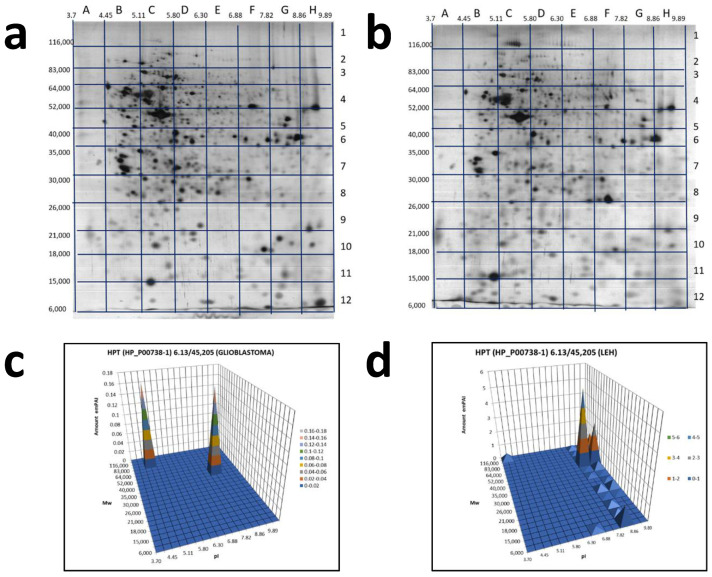
2DE sectional analysis of Hp proteoforms in cells. Glioblastoma 2DE gel (**a**). LEH 2DE gel (**b**). A 2DE pattern of Hp proteoforms in glioblastoma cells (**c**). A 2DE pattern of Hp proteoforms in LEH cells (**d**).

## Data Availability

Not applicable.

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
