# Peer review of "Evaluation of Haptoglobin and Its Proteoforms as Glioblastoma Markers"

_ijms, 2021, doi:10.3390/ijms22126533_

Round 1

Reviewer 1 Report

Dear All,

            Re: ijms-1238365 “Evaluation of Haptoglobin and Its proteoforms as Glioblastoma  markers”

The work in this manuscript explores that the proteoforms of haptoglobin could serve as a putative plasma marker for glioblastoma.  In patients harboring glioblastoma's revealed an up-regulation of alpha2- and beta-chain forms of haptoglobin.  Moreover, glioblastoma patients had a plasma form of un-processed haptoglobin. 

The manuscript requires significant revisions.

  1. For each figure, the subheading of a, b, c, d, etc., should be mentioned in the manuscript's text.

  1. The text refers to figure 6, but figure 6 is not included in the manuscript.Thus, I cannot complete my review of the manuscript until figure 6 is enclosed, or figure 6 does not exist.

Author Response

Dear reviewer, thank you very much for your comments that helped us to improve the manuscript!

Here, our a point-by-point response.

Comments and Suggestions for Authors

For each figure, the subheading of a, b, c, d, etc., should be mentioned in the manuscript's text.

Answer. The subheadings were added.

 The text refers to figure 6, but figure 6 is not included in the manuscript.Thus, I cannot complete my review of the manuscript until figure 6 is enclosed, or figure 6 does not exist.

Answer. Sorry, it was a typo. Of course, it should be Figure 5. But what is interesting, as we made two figures from Figure 1 (Fig1 and Fig2) numeration shifted, and in the corrected manuscript version Figure 5 became Figure 6.

Reviewer 2 Report

In this paper Naryzhny and colleagues investigated the occurrence and state of haptoglobin (Hp) proteoforms both in the plasma of Glioblastoma (GBM) patients and GBM cells, in order to evaluate the use of Hp as a potential GBM-specific biomarker. The introduction is appropriate, with sufficient references.

However, a major point concerns the small number of sample size that the Authors have included in the study, since as stated by Authors in the Conclusions section “The lack of consistency is due to the disease and patient heterogeneity and involvement of these proteins in various physiological and pathological processes. Another important limitation is a wide range of Hp levels in normal serum.” The Authors should take into consideration to implement the number of samples.

Moreover, there are a few points that the Authors should address in order to add consistency to the study.

1-In Figure 1a, it is difficult to readily identify the lanes which corresponds to either GBM or control samples. Instead of using the Arabic number to identify the samples (18, 19, 20…), the Authors should write i.e. “GBM” or “C (control)” in the upper lane of Figure 1a; this would render it clearer to the reader.

2-Moreover, why in the gel stained with Comassie are only reported GBM and control samples, whereas in the statistical analysis of Figure 1b and c are also reported the other groups (i.e. cancer and acute phase)? Lines 97-106, The Authors state that “The representative gel of such analysis is shown in Figure 1”, being this analysis relative to “different groups of plasma samples using SDS_PAGE. These groups were….” (lines 97-101). The Authors should add a representative gel regarding the data and the statistical analysis reported in Figure 1b and c.

3- Lines 111-113: these groups are not reported in Figure 1.

4-Again, why in Figure 1d-f are only reported the WB and the statistical analysis of a2- and b-chains of GBM and control samples, but not of other groups?

5-The Figure legend of Figure 4 is not clear. What does the Authors mean with “a mix of healthy donors’ plasma”?

6- Please check the figure legend of Figure 5. Why the Authors choose LEH human lung fibroblasts? Why the Authors did not use normal human astrocytes, since GBM is a who grade IV astrocytoma?

7- In Materials and Methods section, the Authors reported (lines 270-271): “Also, 16 blood samples… from healthy donors”; but in lines 98-99, the healthy group has n=4;  “other cancers (n=7), c) 98 acute-phase, non-related with cancer (n=5), d) healthy group (n=4).”. Please correct and clarify.

Author Response

Dear reviewer, thank you very much for your comments that helped us to improve the manuscript!

Here, our a point-by-point response

However, a major point concerns the small number of sample size that the Authors have included in the study, since as stated by Authors in the Conclusions section “The lack of consistency is due to the disease and patient heterogeneity and involvement of these proteins in various physiological and pathological processes. Another important limitation is a wide range of Hp levels in normal serum.” The Authors should take into consideration to implement the number of samples.

Answer. We agree that a bigger number of samples can improve the statistics of the data. So we have tried to collect more samples and added them to the analysis.

Moreover, there are a few points that the Authors should address in order to add consistency to the study.

1-In Figure 1a, it is difficult to readily identify the lanes which corresponds to either GBM or control samples. Instead of using the Arabic number to identify the samples (18, 19, 20…), the Authors should write i.e. “GBM” or “C (control)” in the upper lane of Figure 1a; this would render it clearer to the reader.

Answer. We corrected the legends using abbreviations – G (GBM), C (control), T (other cancer), A (acute-phase).

2-Moreover, why in the gel stained with Comassie are only reported GBM and control samples, whereas in the statistical analysis of Figure 1b and c are also reported the other groups (i.e. cancer and acute phase)? Lines 97-106, The Authors state that “The representative gel of such analysis is shown in Figure 1”, being this analysis relative to “different groups of plasma samples using SDS_PAGE. These groups were….” (lines 97-101). The Authors should add a representative gel regarding the data and the statistical analysis reported in Figure 1b and c.

Answer. We added more pictures of the gels

3- Lines 111-113: these groups are not reported in Figure 1.

Answer. We added more gels in the Figure 1

4-Again, why in Figure 1d-f are only reported the WB and the statistical analysis of a2- and b-chains of GBM and control samples, but not of other groups?

Answer. We added more information in the Figure (this is Figure 2 not 1)

5-The Figure legend of Figure 4 is not clear. What does the Authors mean with “a mix of healthy donors’ plasma”?

Answer. We gave an explanation about this sample. This mix is a pooled sample of 54 healthy volunteers’ plasma ( J. Proteome Res. 2016, 15, 11, 4039–4046)

6- Please check the figure legend of Figure 5. Why the Authors choose LEH human lung fibroblasts? Why the Authors did not use normal human astrocytes, since GBM is a who grade IV astrocytoma?

Answer. Unfortunately, we don’t have normal human astrocytes. Sure, it could be nice to have similar data using this cell line, if we want to compare better normal and cancer glial cells. But we represent here the data about haptoglobin in lung cells not as a normal control but as an example of a similar situation with zonulin in glioblastoma and lung.

7- In Materials and Methods section, the Authors reported (lines 270-271): “Also, 16 blood samples… from healthy donors”; but in lines 98-99, the healthy group has n=4;  “other cancers (n=7), c) 98 acute-phase, non-related with cancer (n=5), d) healthy group (n=4).”. Please correct and clarify.

Answer.  The complete sentence you cited is “Also, 16 blood samples were collected in the Pavlov First Saint-Petersburg State Medical University from non-GBM patients and healthy donors”.  So 16 samples are from different groups, not only from healthy people.

These 16 samples are other cancers (n=7), c) acute-phase, non-related with cancer (n=5), d) healthy group (n=4). As we added more samples in the study, we have removed the above sentence.

Round 2

Reviewer 2 Report

The Authors have answered to all the request made by the Reviewer. I have no more academic questions.